# Analysis, Synthesis and Characterization of Thin Films of a-Si:H (n-type and p-type) Deposited by PECVD for Solar Cell Applications

**DOI:** 10.3390/ma14216349

**Published:** 2021-10-24

**Authors:** Abel Garcia-Barrientos, Jose Luis Bernal-Ponce, Jairo Plaza-Castillo, Alberto Cuevas-Salgado, Ariosto Medina-Flores, María Silvia Garcia-Monterrosas, Alfonso Torres-Jacome

**Affiliations:** 1Facultad de Ciencias, Universidad Autónoma de San Luis Potosi, San Luis Potosi 78295, Mexico; 2Departamento de Ingeniería Mecánica, Instituto Tecnológico de Orizaba, Orizaba 94300, Mexico; jberponce@gmail.com (J.L.B.-P.); mgarciam@orizaba.tecnm.mx (M.S.G.-M.); 3Facultad de Ciencias Básicas, Universidad del Atlántico, Barranquilla 081001, Colombia; jaiplaza@gmail.com; 4Departamento de Ingeniería Mecánica, Instituto Tecnológico de Tlalnepantla, Tlalnepantla de Baz 54070, Mexico; ing_cuevass@hotmail.com; 5Instituto de Investigación en Metalurgia y Materiales, Universidad Michoacana de San Nicolás de Hidalgo, Morelia 58030, Mexico; ariosto@umich.mx; 6Departamento de Electrónica, Instituto Nacional de Astrofísica, Óptica y Electrónica (INAOE), Tonantzintla 72840, Mexico; atorres@inaoep.mx

**Keywords:** a-Si:H, PECVD, solar cells

## Abstract

In this paper, the analysis, synthesis and characterization of thin films of a-Si:H deposited by PECVD were carried out. Three types of films were deposited: In the first series (00 process), an intrinsic a-Si:H film was doped. In the second series (A1–A5 process), n-type samples were doped, and to carry this out, a gas mixture of silane (SiH_4_), dihydrogen (H_2_) and phosphine (PH_3_) was used. In the third series (B1–B5 process), p-type samples were doped using a mixture of silane (SiH_4_), dihydrogen (H_2_) and diborane (B_2_H_6_)_._ The films’ surface morphology was characterized by atomic force microscopy (AFM), while the analysis of the films was performed by scanning electron microscopy (SEM), and UV–visible ellipsometry was used to obtain the optical band gap and film thickness. According to the results of the present study, it can be concluded that the best conditions can be obtained when the flow of dopant gases (phosphine or diborane) increases, as seen in the conductivity graphs, where the films with the highest flow of dopant gas reached the highest conductivities compared to the minimum required for materials made of a-Si:H silicon for high-quality solar cells. It can be concluded from the results that the magnitude of the conductivity, which increased by several orders, represents an important result, since we could improve the efficiency of solar cells based on a-Si:H.

## 1. Introduction

The principal task to diminish the disturbance of energetic needs due to the nature of humans is the generation of renewable energy. Additionally, energy consumption is growing steeply and is directly proportional to global population growth. The high efficiency of solar cells has always been sought after for effective light management. Today, the manufacture of materials to produce high-efficiency solar cells has been a key factor in the development of innovative electrical devices. In recent decades, amorphous silicon (a-Si) has been an innovative material in the microelectronic industry, due to its fabrication being cheap in comparison with that of crystalline silicon (c–Si). Amorphous silicon is the non-crystalline form of silicon, and it has a band gap of around 1.7 eV. Hydrogenated amorphous silicon (a-Si:H) materials have received a great deal of attention for their potential to produce inexpensive solar cells, particularly in the polycrystalline growth regime. In this material, the atoms are not located at a specific distance or specific angles; therefore, it has a random network. In [1], it was indicated that in hydrogen added to amorphous silicon, it is possible to find a beneficial effect. Thus, [2,3] showed that silicon has semiconducting properties when combined with a dopant gas such as phosphine or diborane. Amorphous silicon has been deposited through various techniques such as sputtering, and, with better results, plasma enhanced chemical vapor deposition (PECVD) has been implemented since, with this technique, it is possible to adjust the optoelectronic properties to the variation in the ambient temperature, the RF frequency and some other parameters of the process [3,4,5]. However, a-Si:H films have several drawbacks such as poor electron transport properties, low carrier mobility, a large number of defects and poor stability against radiation, which limit their applications in new and high-performance devices. In [6], the evolution of the a-Si:H to nc-Si:H transition of hydrogenated silicon films deposited by trichlorosilane was presented. In this aspect, there are some studies that have explored the optimization of the deposition conditions of a-Si:H films by the PECVD technique, where the goal is to improve its performance characteristics and to produce other materials with different structural, electrical and optical properties [7,8,9,10]. The preparation and measurement of highly efficient a-Si:H single-junction solar cells were presented in [11,12,13], and the impact of the deposition rate on metastability was discussed in [14]. The practical applications of building integrated photovoltaic systems were presented in [15,16,17,18,19,20]. Moreover, in [21], plasmonic light trapping phenomena in thin-film silicon solar cells with improved self-assembled silver nanoparticles were discussed. Additionally, high-efficiency a-Si:H/μc-Si:H solar cells by optimizing a-Si:H and μc-Si:H sub-cells were presented in [22], and the optical modeling of a-Si:H-based solar cells on textured substrates was discussed in [23,24]. Similarly, in [25], a study on n- and p-type doping of PECVD a-SiC:H obtained under silane starving plasma conditions with and without hydrogen dilution was presented. Some recent research reports [26,27] have provided a complete and comprehensive analysis of the thin-film amorphous silicon solar cell market focusing on revenue, growth patterns, market trends and the overall volume, along with a detailed analysis of all of the significant factors affecting the marketplace development of the industry and demand forecast to 2027. Moreover, a review on approaches to improve the power output of building integrated photovoltaic systems was presented in [28,29,30,31]. Therefore, in this research work, the analysis, synthesis and characterization of thin films of a-Si:H (n-type and p-type) deposited by PECVD were carried out, where the objective of this study was to find an optimized semitransparent a-Si:H thin film for solar cell applications. This optimization was analyzed by a-Si:H thin films that have been doped with phosphane (PH_3_) (n-type) and with diborane (B_2_H_6_) (p-type). The procedure was repeated with different values for the flow of PH_3_ and H_2_ for n-type films and B_2_H_6_ and H_2_ for p-type films.

## 2. Materials and Methods

Three types of films were deposited: an intrinsic a-Si:H thin film (00 process), called the first series, and two series of a-Si:H films doped by the PECVD technique. In the second series (A1–A5 process), n-type samples were doped using a gas mixture of SiH_4_, H_2_ and PH_3_. In the third series, p-type films were doped using a mixture of SiH_4_, H_2_ and B_2_H_6_ (B1–B6 process)_._ In both series, samples were made of glass (Corning Glass 2974 and 1737). We used the AMP-3300 Plasma II PECVD Deposition System. The invariant deposition conditions for series A and B were ambient pressure 600 mTorr, a power of 300 W, power density of 90 mW/cm^2^, temperature of 300 °C and 30 min treatment time. Flows of H_2_, SiH_4_ and PH_3_ gases in the second series or H_2_, SiH_4_ and B_2_H_6_ in the third series were adjusted according to some requirements, as shown in Table 1.

The thickness and band gap energy of the samples in the three series are shown in Table 2.

Sample characterization and process evaluation were performed by the measurements of absorption coefficients, the conductivities, the activation energies and the thicknesses of the films. Ultraviolet–visible (UV–Vis) transmittance measurements were performed in a Perkin-Elmer Lambda 3B spectrometer (Boston, MA, USA) in the range of 300–900 nm. Activation energy and conductivities were obtained by using measurements of current–voltage in a wide range of temperatures (300 to 400 K). For the electrical characterization, a micro-spike system Model LTMP-2, “MMR Technology Inc” (San Jose, CA, USA), was used. Spectroscopic ellipsometry was used to obtain the measurements of the imaginary part of the pseudo-dielectric function of the films, data which were useful to obtain their thicknesses. For this, a Jobin Yvon ellipsometer-MWR UVISEL UV/Vis (Villeneuve-d’Ascq, France) was used.

## 3. Results

Figure 1 shows the FTIR spectra for a-Si:H in the second and third series of a-Si:H thin films (Figure 1a shows n-type films with different processes, A1–A5, and Figure 1b shows p-type films with different processes, B1–B5). The spectrum of a-Si:H consisted of typical absorption bands for different types of hydrogen bonding to nitrogen and silicon and silicon−nitrogen-related bands. One can observe a peak around 640 cm^−1^, which corresponds to Si–H rocking/waging modes, and the band around 800–950 cm^−1^ is related to SiH_2_ or SiH_3_ bonding with low intensity. The main trend is that the intensity of the bands assigned to SiHn decreases with the increase in the ammonia content in PECVD. The spectra of transmittance with respect to the wavelength of the doped n-type films are shown in Figure 2a, and in Figure 2b for the p-type films. Here, it is observed that, in the range from 300 to 550 nm, the intrinsic film shows a greater transmittance. However, from 550 to 900 nm, the doped films, either n-type or p-type, show almost double the transmittance in comparison to the intrinsic film. The refractive index with respect to the wavelength is shown in Figure 3 (for doped n-type films in Figure 3a, and for p-type films in Figure 3b). It can be seen, in regions of the spectrum where the material does not absorb light, that the refractive index tends to decrease with increasing wavelength. The absorption coefficient vs. phonon energy for the two series of thin films: (a) n-type films with different processes, A1–A5, and (b) p-type films with different processes, B1–B5, is shown in Figure 4a,b, respectively. In Figure 5, one can see the relation between conductivity and 1/kT for the two series of thin films (in Figure 5a for the n-type films, and in Figure 5b for the p-type films). In general, it can be observed that the deposited films with less flow of H_2_ (200 mTorr), for both doped series, exhibit the opposite behavior in each type of doped film; this means, while the n-type film shows a greater transmittance at a greater wavelength, the p-type film shows little transmittance at a great wavelength, the latter having a similar behavior to the one observed in process 00 (intrinsic film).

In Figure 5a, one can observe that for A1 (H_2_ = 200 sccm), the conductivity suddenly dropped. This can be explained by the effect of the hydrogen content in the a-Si:H thin films. It is caused by the deteriorated interface and can be attributed to voids created by insufficiently passivated Si surface dangling bonds, voids formed by SiH_2_ clusters and voids formed by Si particles caused by gas-phase particle formation in silane plasma [30].

## 4. Discussion

In this study, n- and p-type a-Si:H thin films were deposited by the PECVD technique at a low frequency, varying the fluxes of the diborane and phosphine gases. When comparing the intrinsic sample with the doped samples, it can be noted that the doped samples have better properties than the intrinsic sample; this fact is evidenced in the conductivity graph where the doped samples show a higher conductivity, and in the fact that their gap improves at a higher dopant flux. From this, it can also be concluded that the best conditions are presented with the highest fluxes of dopants, either phosphine gas or diborane gas. In this way, the conductivity graph shows samples with the highest flow of dopants reaching conductivities higher than the minimum conductivity required in a-Si: H materials for quality solar cells.

As for which of the doped films (type p or type n) is better for the manufacture of solar cells, it is concluded that the A3 and B3 processes present the best results, which can be seen in the transmittance and absorption coefficient curves. Finally, if we analyze how the hydrogen flux during deposition influences the properties of the films, it can be concluded that the deposited films with a lower hydrogen flux (200 sccm) do not present a greater attractiveness than the samples deposited with a higher hydrogen flux (1000 sccm), since the latter have better optical and electrical properties. This can be observed in the conductivity curve.

In Figure 6, five images obtained by AFM are shown, corresponding to the morphology of the a-Si:H films deposited over the glass substrate with flows of H_2_ = 1000 sccm and SiH_4_ = 50 sccm. In the image, it can be observed that for a pressure of 600 mTorr, there is no evidence of nanoclusters, and they appear from 1000 mTorr, where their quantity increases with the increase in pressure, but their size decreases when the pressure increases. A similar behavior, in a greater quantity of nanoclusters, is seen in the films shown in Figure 7. Figure 8 shows the SEM photo of the sample at 2000 mTorr, where nano-crystals appear that are approximately 1.5 nm in size.

## 5. Conclusions

In this study, n-type and p-type a-Si:H films were successfully deposited by using the PECVD technique at a low frequency, always changing the flows of the diborane and phosphine gases. When comparing the intrinsic sample with the doped films, it can be seen that the doped films present better electric properties than the intrinsic film. This can be evidenced in the graphs of conductivity, where the doped films present a greater conductivity, and by the fact that their gap becomes better with a greater flow of dopant gases. It can also be concluded that the best conditions can be obtained using greater flows of the dopant gas, this being either phosphine or diborane. It was found that for our deposition system and conditions, processes A3 and B3 presented the best optically optimized a-Si:H thin films, due to the transmittance curve. However, processes A5 and B5 presented better results with respect to their conductivity. With this research, it is confirmed that films of amorphous silicon with controlled chamber pressures and gas flows deposited by the PECVD technique can be obtained. The influence of H_2_ and SiH_4_ in the deposition of a-Si:H thin films by the PECVD process can improve the efficiency of solar cells.

## Figures and Tables

**Figure 1 materials-14-06349-f001:**
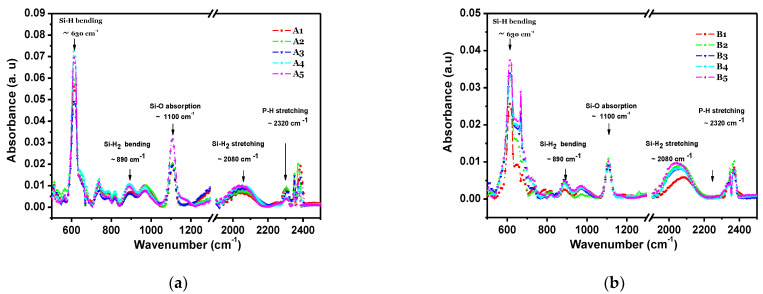
FTIR spectra for a−Si:H for two series of thin films: (**a**) n−type films with different H_2_ and PH_3_ contents; (**b**) p−type films with different H_2_ and B_2_H_6_ contents.

**Figure 2 materials-14-06349-f002:**
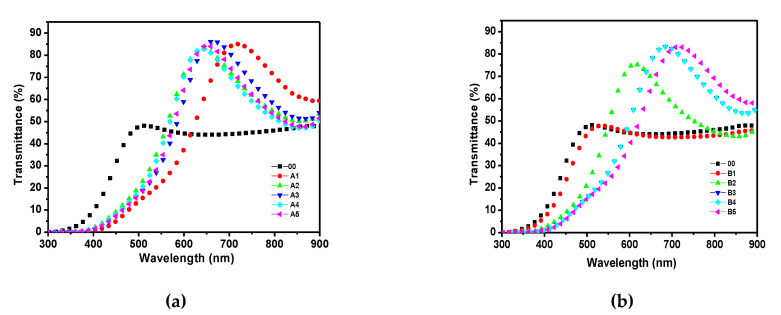
Transmittance vs. wavelength for two series of thin films: (**a**) n-type films with different H_2_ and PH_3_ contents; (**b**) p-type films with different H_2_ and B_2_H_6_ contents.

**Figure 3 materials-14-06349-f003:**
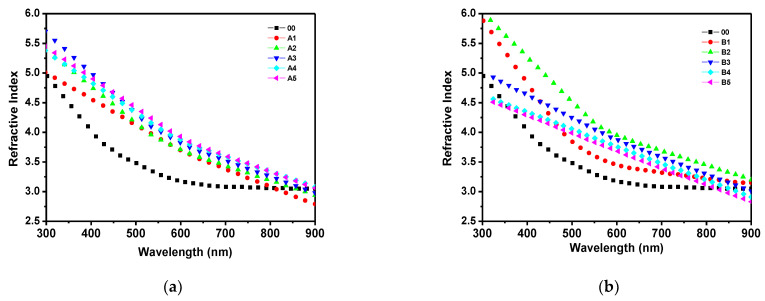
Refractive index vs. wavelength for two series of thin films: (**a**) n-type films with different H_2_ and PH_3_ contents; (**b**) p-type films with different H_2_ and B_2_H_6_ contents.

**Figure 4 materials-14-06349-f004:**
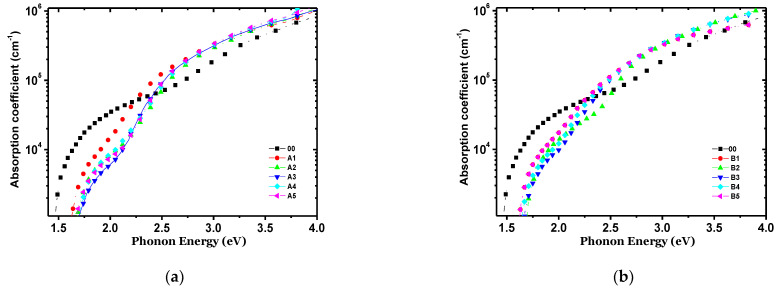
Absorption coefficient vs. phonon energy for two series of thin films: (**a**) n−type films with different H_2_ and PH_3_ contents; (**b**) p−type films with different H_2_ and B_2_H_6_ contents.

**Figure 5 materials-14-06349-f005:**
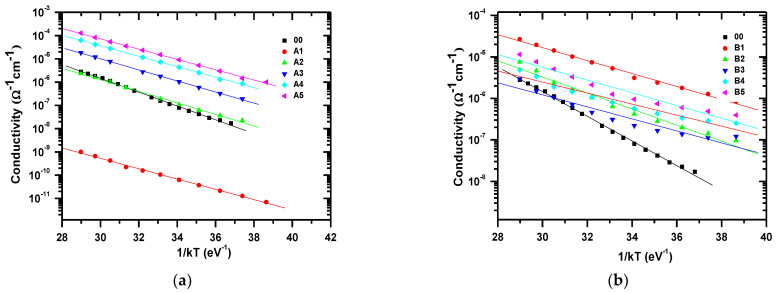
Relation between conductivity and 1/kT for two series of thin films: (**a**) n−type films with different H_2_ and PH_3_ contents; (**b**) p−type films with different H_2_ and B_2_H_6_ contents.

**Figure 6 materials-14-06349-f006:**
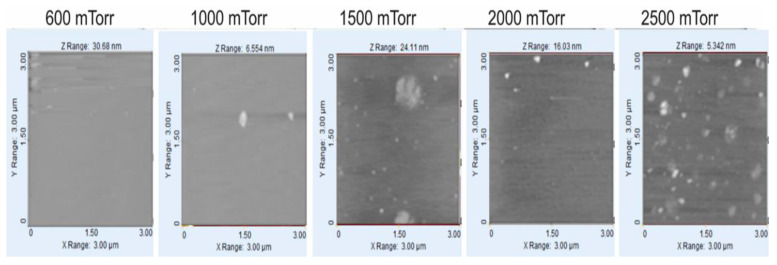
Images of surface of a-Si:H films deposited over glass substrate with flows of H_2_ = 1000 sccm and SiH_4_ = 50 sccm.

**Figure 7 materials-14-06349-f007:**
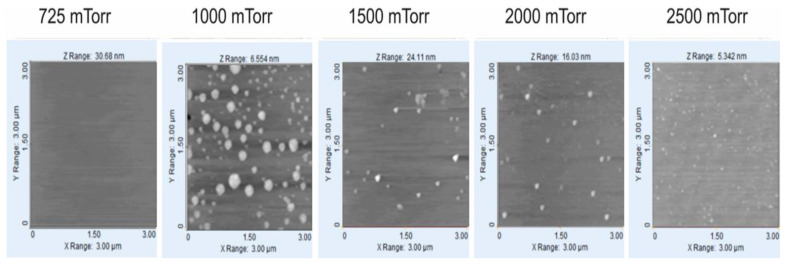
Images of surface of a-Si:H films deposited over glass substrate with flows of H_2_ = 4000 sccm and SiH_4_ = 200 sccm.

**Figure 8 materials-14-06349-f008:**
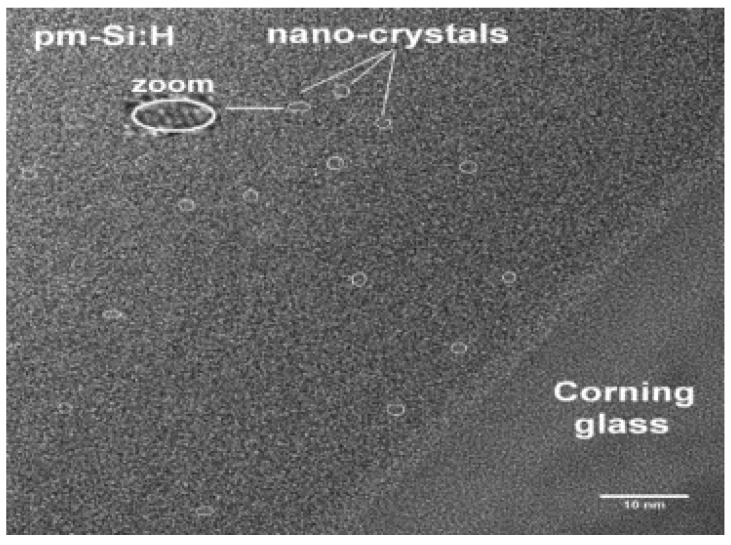
SEM photo of the sample at 2000 mTorr, where nano-crystals appear.

**Table 1 materials-14-06349-t001:** Gas flows of H_2_, SiH_4_, PH_3_ and B_2_H_6_ for the first, second and third series.

Process ^1^	H_2_ (sccm)	SiH_4_ (sccm)	PH_3_ (sccm)	B_2_H_6_ (sccm)
00	1000	50	0	0
A1	200	50	4	0
A2	1000	50	5	0
A3	1000	50	6	0
A4	1000	50	8	0
A5	1000	50	10	0
B1	200	50	0	4
B2	1000	50	0	5
B3	1000	50	0	6
B4	1000	50	0	8
B5	1000	50	0	10

^1^ Pressure 600 mTorr, power 300 W, power density 90 mW/cm^2^, time 30 min and temperature 300 °C.

**Table 2 materials-14-06349-t002:** Thickness and energy gap of the three series.

Process	Thickness (Å)	Eg (eV)
00	583	1.97
A1	1066	1.71
A2	941	1.71
A3	995	1.82
A4	910	1.79
A5	955	1.78
B1	601	1.83
B2	860	1.74
B3	985	1.70
B4	1030	1.69
B5	1079	1.67

## Data Availability

Data are contained within this article.

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
