# Peer review of "Analysis, Synthesis and Characterization of Thin Films of a-Si:H (n-type and p-type) Deposited by PECVD for Solar Cell Applications"

_materials, 2021, doi:10.3390/ma14216349_

Round 1
Reviewer 1 Report
Comments:
The author presented the synthesis and characterization of a-Si:H thin films prepared by the PECVD method. They deposited three types of films:
- Intrinsic a-Si:H film (called control sample).
- N-type doped a-Si: H film, and it is done by a mixture of SiH4, H2 and PH3 gases.
- P-type doped a-Si: H film, and it is done by a mixture of SiH4, H2 and B2H6 gases.
These deposited films were characterized by AFM and TEM (for morphology), UV-Vis ellipsometry (to obtain bandgap and film thickness)
The manuscript is in order, and I have the following comments.
- There are many terms that are in short form, and the author needs to write their full forms. I am just maintaining a few like SiH4, H2, and PH3, AFM, TEM, PECVD, …etc; and the author also needs to find all.
- The similar work has been reported by A.R. Oliveira et al. (Materials Science and Engineering B 128 (2006) 44–49, N and p-type doping of PECVD a-SiC:H obtained under “silane starving plasma” condition with and without hydrogen dilution). You may have to cover it in the introduction.
- In the materials and methods section, the author mentioned the second series (page 89) and the third series (page 91), and it could be good if the author mentioned the first series name.
- In table 2: the thickness is in angstrom, right? I think the symbol need to change. It should be Å.
- Figure (2) shows the transmittance spectra of second and third series films. In both cases (A1-A5) and (B2-B5) their transmittance is more than 70 % between (600- 800 nm), and for solar cell materials requirement, the transmittance should be low in the visible region. So, how you proposed that the doped material is effective for solar cells application?
- In Figure (5a), for A1, the conductivity suddenly dropped by 10E-4 fold; why? I think it would be good if the author could put some content in the main text.
- In Figure (6) (AFM images), it could be good to mention the scale bar and surface roughness value as they can be easily calculated from AFM data. If possible, please put the z-scale. Also, how do those nanocluster sizes affect the film optical and electrical properties?
- Figures (7) look like AFM images, right? Auther mentioned in the abstract that they have also shown the TEM images. I did not find the TEM image in the manuscript. Please check.
- In the conclusion section, it could be attractive if the author could mention their outcomes, like which film condition (A1-A5 and B1-B5) delivered better results.
Author Response
Dear Reviewer,
Thank you very much for your comments. They were really excellent, and they help us to present the paper in the best form. Please, if you have any more comments, tell me as soon as possible.
Best regards
Response to Reviewer 1 Comments
The author presented the synthesis and characterization of a-Si:H thin films prepared by the PECVD method. They deposited three types of films:
- Intrinsic a-Si:H film (called control sample).
- N-type doped a-Si: H film, and it is done by a mixture of SiH4, H2 and PH3 gases.
- P-type doped a-Si: H film, and it is done by a mixture of SiH4, H2 and B2H6 gases.
These deposited films were characterized by AFM and TEM (for morphology), UV-Vis ellipsometry (to obtain bandgap and film thickness)
The manuscript is in order, and I have the following comments.
- There are many terms that are in short form, and the author needs to write their full forms. I am just maintaining a few like SiH4, H2, and PH3, AFM, TEM, PECVD, …etc; and the author also needs to find all.
Response 1: These terms already were written in their full form.
- The similar work has been reported by A.R. Oliveira et al. (Materials Science and Engineering B 128 (2006) 44–49, N and p-type doping of PECVD a-SiC:H obtained under “silane starving plasma” condition with and without hydrogen dilution). You may have to cover it in the introduction.
Response 2: This paper already included in the introduction.
- In the materials and methods section, the author mentioned the second series (page 89) and the third series (page 91), and it could be good if the author mentioned the first series name.
Response 3: This already was mentioned.
- In table 2: the thickness is in angstrom, right? I think the symbol need to change. It should be Å.
Response 4: This was changed.
- Figure (2) shows the transmittance spectra of second and third series films. In both cases (A1-A5) and (B2-B5) their transmittance is more than 70 % between (600- 800 nm), and for solar cell materials requirement, the transmittance should be low in the visible region. So, how you proposed that the doped material is effective for solar cells application?
Response 5: The curve in figure 2 is typically for solar cells applications. For example, the figure below describes the total transmission of an optimized semi-transparent solar cell and that of the standard semi-transparent solar cell. The transmission was enhanced in the 600–700 nm wavelength range.
- Brodu, C. Seydoux, G. Finazzi, C. Dublanche-Tixier, C. Ducros. Optical optimization of semitransparent a-Si:H solar cells for photobioreactor application. Thin Solid Films, Elsevier, 2019, 689, pp.137492. ff10.1016/j.tsf.2019.137492ff. ffhal-02332917f
- In Figure (5a), for A1, the conductivity suddenly dropped by 10E-4 fold; why? I think it would be good if the author could put some content in the main text.
Response 6: This was included in lines 177-181.
- In Figure (6) (AFM images), it could be good to mention the scale bar and surface roughness value as they can be easily calculated from AFM data. If possible, please put the z-scale. Also, how do those nanocluster sizes affect the film optical and electrical properties?
Response 7: It is a good question. We are sure that the nanocluster size affects the optical and electrical properties the thin film a-Si:H but how exactly?, to be honest we need to do a new experiments and characterizations and try to find a relation between nanocluster sizes vs electrical and optical properties. It is a good point. I promise I will do it soon.
- Figures (7) look like AFM images, right? Auther mentioned in the abstract that they have also shown the TEM images. I did not find the TEM image in the manuscript. Please check.
Response 8: We include the Figure 8.
- In the conclusion section, it could be attractive if the author could mention their outcomes, like which film condition (A1-A5 and B1-B5) delivered better results.
Response 9: This was included in the conclusions, lines 242-245.
Reviewer 2 Report
The article submitted in Materials requires major revision. I have the following major comments that must be addressed before final acceptance.
- There are several typos and grammatical errors. The technical language of the manuscript needs thorough revision. Please take the help of a native speaker.
- There is a large difference in the energy band values from process (00) to B5 (Table 1). How does this difference influence the conductivities of the films?
- In Figure 2, there is a significant difference in the transmittance of intrinsic films with those of doped films with respect to specific wavelength regions. What is the physical reason for this?
- Please check the vertical axis of Figure 1 and Figure 4. Is it correct to choose absorption coefficients with or without a unit?
- In a paragraph (lines #172-183), it is a quite confusing explanation about the impact of hydrogen flux on the properties of thin films. Under what conditions authors decided which flux is better for which type of doping. Please rewrite.
- I believe that results and discussion can be combined. Also, please try to concise the conclusions. It is quite redundant.
- Please refer to recent studies of solar cells to enhance the introduction. (https://doi.org/10.1016/j.apsusc.2021.150852; https://doi.org/10.3390/en14010178;)
Author Response
Dear Reviewer,
Thank you very much for your comments. They were really excellent, and they help us to present the paper in the best form. Please, if you have any more comments, tell us as soon as possible.
Best regards
Response to Reviewer 2 Comments
The article submitted in Materials requires major revision. I have the following major comments that must be addressed before final acceptance.
- There are several typos and grammatical errors. The technical language of the manuscript needs thorough revision. Please take the help of a native speaker.
Response 1: We rewrite the introduction, and we make a revision with help of a native speaker.
- There is a large difference in the energy band values from process (00) to B5 (Table 1). How does this difference influence the conductivities of the films?
Response 2:
Significant correlation was found between the electrical conductivity and the optical gap values in a-Si:H thin films. It can be observed in Figure 5b. For it is predominantly influenced by the transition tail width of undoped and doped a-Si:H thin films. Our material (a-Si:H) is a semiconductor material in which the band structure is characterized by smooth variation of the density of states with energy in the band-edge zones, called band tails, and a high density of states in the midgap region. If the mobility of carriers in the band tails is high enough, the conduction mechanism is dominated by carriers activated from the midgap states to these band tails.
- In Figure 2, there is a significant difference in the transmittance of intrinsic films with those of doped films with respect to specific wavelength regions. What is the physical reason for this?
Response 3: This observation could be attributed to the effect quantum confinement as a result of the reduced particle dimension as the dopants tend to blend with the host atoms.
- Please check the vertical axis of Figure 1 and Figure 4. Is it correct to choose absorption coefficients with or without a unit?
Response 4: It was revised and corrected.
- In a paragraph (lines #172-183), it is a quite confusing explanation about the impact of hydrogen flux on the properties of thin films. Under what conditions authors decided which flux is better for which type of doping. Please rewrite.
Response 5: This was rewritten in lines 192-199.
- I believe that results and discussion can be combined. Also, please try to concise the conclusions. It is quite redundant.
Response 6: These were rewritten and revised.
- Please refer to recent studies of solar cells to enhance the introduction. (https://doi.org/10.1016/j.apsusc.2021.150852; https://doi.org/10.3390/en14010178;)
Response 7: These references were included in the introduction
Reviewer 3 Report
There are a lot of problems in the paper that must be corrected:
1) There is no clear idea of the work that is presented in the manuscript. From the introduction section its not clear what problem was solved by the work presented in the paper or what new idea is presented that leads to a new findings for the science in this field.This must be corrected.
2) Information presented in the introduction section is quite old and does not represent latest data in this field.This must be corrected.
3) There is no information about equipment that was used for sample fabrication.This must be corrected.
4) There are some misused physical terms like "potency density" add etc. This must be corrected.
5) As there is no idea of the paper, so the conclusions also don't represent any new finding or new ideas. Also it is not clear if the goals of the paper are achieved as the goals are not present.
Author Response
Dear Reviewer,
Thank you very much for your comments. They were really excellent, and they help us to present the paper in the best form. Please, if you have any more comments, tell us as soon as possible.
Best regards

Round 2
Reviewer 1 Report
Comments
The author resolved most of the comments, and there are a few things that remain.
1. In an abstract, a few terms like SiH4, H2 and PH3, AFM still remain and need to be put the full form. These may be standard terms, but they could be difficult for new readers or readers from different domains to understand.
2. In the abstract, the author still mentioned the TEM details, but the main manuscript texts do not have the same. I think in place of TEM author mentioned the SEM image. In that case, the author needs to correct it in an abstract section.
3. In an AFM image, the still scale bar is missing. Please check it.
Author Response
Dear Reviewer,
Thank you very much for your comments again. They were excellent, and they help us to present the paper in the best form. Please, if you have any more comments, tell us as soon as possible.
Best regards

Reviewer 2 Report
Overall, the paper has been significantly improved and it can be accepted after incorporating the minor suggestions for improvement. Please check the references are properly cited because in some references there are full authors included with their full names while in some abbreviations of their names were used. The same type of issue with the volume, issue, and page numbers in the references such as ref#10,19,29, 30. Please be consistent in citing all the references according to the journal style guide. Also, please check a recent study (Jung-Hoon Lee et al, NPG Asia Mater 13, 43, 2021)
Author Response
Dear Reviewer,
Thank you very much for your comments again. They were excellent too, and they help us to present the paper in the best form. Please, if you have any more comments, tell us as soon as possible.
Best regards

Round 3
Reviewer 1 Report
My comments are as follows:
The author resolved all comments, and I think I still have two minor comments.
1. In the abstract, there is one sentence “as well UV-Visible Ellipsometry was used to obtain the optical band gap, films thickness and deposition rate” (Line number 20 and 21). How the deposition rate can be calculated from ellipsometry? Please check it and correct the sentence.
2. Author mentioned in an abstract that the cross-section of the film was performed by Scanning Electron Microscopy (SEM), and the image shown in Figure (7) does not seem like a cross-section image. Please check and correct the sentence or image.
Author Response
Dear Reviewer,
Thank you so much for your comments again. They were excellent, and they help us to present the paper in the best form. Please, if you have any more comments, tell us as soon as possible.
Best regards
